# Hallucinating for Diagnosing: One-Shot Medical Image Classification Leveraging Score-Based Generative Models

**Eva Pachetti**[1,2]                                                        EVA.PACHETTI@ISTI.CNR.IT

**Sara Colantonio**[1]                                              SARA.COLANTONIO@ISTI.CNR.IT

[1] *"Alessandro Faedo" Institute of Information Science and Technologies (ISTI), Pisa, Italy*

[2] *Department of Information Engineering, University of Pisa, Pisa, Italy* and

**Editors:** Accepted for publication at MIDL 2024

## Abstract

Deep learning models in data-scarce domains, such as medical imaging, often suffer from poor performance due to the challenges of acquiring large amounts of labeled data. Few-shot learning offers a promising solution to this problem. This work proposes a novel framework to jointly train a score-based generative model for high-quality sample hallucination and a meta-learning framework for one-shot classification. We evaluate our approach on MRI scans of prostate cancer, aiming to classify tumors based on severity. Our preliminary experiments demonstrate promising results, indicating the efficacy of our proposed method in improving classification performance. Future work will involve further analysis using a diverse set of score models and prototypical meta-learning techniques, as well as evaluation of the effectiveness of our framework in other medical imaging tasks.

**Keywords:** One-shot learning, Score-based generative models, Medical image classification

## 1. Introduction

Due to limited data availability, deep learning models often struggle in real-world clinical applications. This limited data restricts the accuracy of trained models, interfering with their practical use. Few-shot learning (FSL) has emerged as a promising approach to address training challenges in data-scarce domains like medical imaging. FSL benefits significantly from meta-learning, which aims to enhance a model's ability to learn robust features by training the model on a set of tasks, often called episodes. A popular meta-learning technique utilizes a prototype-based approach. In each episode, a prototype for each class is created based on the available support examples. Classification then occurs by measuring the distance between a query sample and each class prototype.

Several strategies have been proposed to create more informative prototypes in FSL, such as performing a weighted prototype estimation (Cao et al., 2023), optimizing prototype features extraction (He et al., 2023), and using prior knowledge to perform prototype completion (Zhang et al., 2021). Within the realm of generative models, Zhang et al. (Zhang et al., 2023) proposed a method to generate more informative prototypes by hallucinating additional support samples. Their approach relies on a Variational Autoencoder (VAE) to model the inter-sample difference distribution. However, VAEs often struggle with capturing the full complexity of images due to their reliance on a Gaussian-distributed latent space, leading to blurry or imprecise image generation and, thus, to less meaningful prototypes.

This work addresses a one-shot learning (OSL) classification task by leveraging score-based generative models (SGMs) and prototypical meta-learning. SGMs, unlike VAEs, can infer the actual underlying data distribution by estimating the score function, i.e., the gradient of the log probability density w.r.t. data (Song et al., 2020). This capability allows for generating more realistic and informative samples. Specifically, we propose a novel joint-training approach within an episodic framework. During each episode, the support samples (one for each class) are used to train the SGM, which generates additional samples for each class. The generated and actual support samples are fed into a feature extractor for meta-training. The combined use of actual and synthetic data fosters the creation of more meaningful prototypes, leading to enhanced classification accuracy.

## 2. Methods

### 2.1. Proposal

We leveraged the Meta Deep Brownian Distance Covariance (Meta DeepBDC) (Xie et al., 2022) as a prototypical meta-learning framework. Meta DeepBDC is a powerful prototypical-based model that utilizes the BDC metric to capture more meaningful relationships between embeddings. Specifically, the idea is to compute a BDC matrix for each support and query embedding and calculate the prototype for each class as the average of the support BDC matrices of that class. On the other hand, to generate additional support samples, we leveraged a Denoising Diffusion Probabilistic Model (DDPM) (Sohl-Dickstein et al., 2015; Ho et al., 2020). Traditional DDPMs corrupt the data using a finite number of noise steps and train a probabilistic model sequence to reverse each noise corruption step. In this work, we adopted the approach proposed by Song et al. (Song et al., 2020), which utilizes a continuous noise-perturbation process modeled by stochastic differential equations (SDEs).

To accelerate the convergence of the DDPM during training, we employed a two-step approach. First, we pre-trained the DDPM on an unlabeled dataset to provide it with prior knowledge. In the second phase, we performed joint episodic training between a feature extractor and the generative model. During each episode, the DDPM is trained using the support sample of each class. Then, exploiting the class embedding, it performs a conditional generation of additional support data following the approach of Ho and Salimans (Ho and Salimans, 2021). The feature extractor computes the embedding of actual and generated samples, and their BDC matrix is calculated. The class prototype is then obtained as the average of that class's BDC matrices. Eventually, classification is performed by measuring the cosine distance between the query samples and the class prototypes.

### 2.2. Dataset

We evaluated our approach on multiparametric MRI data of prostate cancer from the PI-CAI dataset (Saha et al., 2023). Our experiments focused on T2-weighted images from patients with both cancerous and benign lesions. We used all benign lesion images (11202) as an unlabeled dataset to perform the DDPM pre-training. In contrast, we leveraged the images of cancerous lesions (2049) for the supervised mets-training phase, dividing them into meta-training (1611), meta-validation (200), and meta-test sets (238). Each acquisition was assigned two ground truth labels: the Gleason Score (GS), indicating the lesion severity,

which ranges from $3 + 4$ to $5 + 5$, reflecting increasing aggressiveness, and the GS's group affiliation, a prognostic score defined by the International Society of Urological Pathology (ISUP) (Egevad et al., 2016), ranging from 2 to 5.

### 2.3. Experiments

We aimed to classify prostate cancer images according to the lesion aggressiveness. Specifically, we tackled two distinct tasks: a *4-way* classification task based on the ISUP value (ranging from 2 to 5) and a *2-way* task, i.e., distinguishing into low-grade (LG) lesions (ISUP $\leq 2$) and high-grade (HG) lesions (ISUP $\geq 3$). For the *4-way* task, we conducted two separate experiments: one involved meta-training and meta-testing using the same set of classes (ISUP 2-5), an approach also taken for the *2-way* task, while the other involved meta-training on the GS classes and meta-testing on ISUP classes. To distinguish between these two approaches, we denote the former scenario as *coarse* and the latter as *fine*. We evaluated the performance of our classification tasks as the mean and standard deviation of the Area Under the Receiver Operating Characteristic Curve (AUROC) across ten meta-testing episodes.

We employed a ResNet-18 architecture as the feature extractor and trained it with the generative model for 100 epochs. Each epoch included ten training and validation episodes. The hyperparameters for our classification backbone included a learning rate of $10^{-4}$, weight decay of $10^{-2}$, AUC margin loss (Yuan et al., 2021), and the Proximal Epoch Stochastic method as the optimizer (Guo et al., 2023). For the DDPM, we employed a learning rate of $20^{-4}$ and Adam as the optimizer. To corrupt the images, we considered 1000 noise scales spanning from 0.1 to 20 variance according to a variance-preserving SDE.

## 3. Results and Conclusions

As shown in Table 1, our preliminary results demonstrate that generating additional support samples improves the mean AUROC across meta-test episodes, suggesting that including synthetic samples leads to more informative prototypes for each class. Future work will investigate further to improve our approach and extend its applicability to other prototypical frameworks and medical imaging classification tasks.

Table 1: Summary of results.

| Task | Baseline | +1 support | +2 support |
|------|----------|-----------|-----------|
| *2-way (coarse)* | 0.504 (0.154) | 0.508 (0.187) | **0.512 (0.204)** |
| *4-way (coarse)* | 0.554 (0.068) | 0.569 (0.036) | **0.570 (0.036)** |
| *4-way (fine)* | 0.579 (0.061) | 0.552 (0.048) | **0.631 (0.078)** |

### Acknowledgments

This study was funded by the European Union's Horizon 2020 Research and Innovation Program under Grant Agreement No 952159 (ProCAncer-I) and by the Regional Project PAR FAS Tuscany-NAVIGATOR.

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
