# OpenReview forum: "Hallucinating for Diagnosing: One-Shot Medical Image Classification Leveraging Score-Based Generative Models"
_MIDL.io/2024/Short_Papers — MIDL 2024 Short Papers_

### Official Review · Reviewer_j2Bu · 2024-04-25

**Confidence:** 4
**Final Rating:** 3.5

**Review:**

The paper "Hallucinating for Diagnosing: One-Shot Medical Image Classification Leveraging Score-Based Generative Models" presents a novel framework that combines score-based generative models with meta-learning for one-shot medical image classification, focusing on MRI scans of prostate cancer. The paper addresses the challenge of data scarcity in medical imaging by leveraging few-shot learning techniques to improve classification performance.

Merits:
Innovative Approach: The integration of score-based generative models with meta-learning for one-shot classification is a novel and promising approach in the field of medical imaging.
Promising Results: The preliminary experiments demonstrate promising results, indicating the efficacy of the proposed method in improving classification performance, which is crucial for accurate diagnosis in medical settings.
Relevance: The application of the framework to classify tumors based on severity in MRI scans of prostate cancer addresses a significant and clinically relevant problem in healthcare.

Limitations:
Limited Evaluation: The paper mentions preliminary experiments with promising results, but a more comprehensive evaluation with larger datasets and comparison to existing methods would strengthen the validity of the proposed framework.
Generalizability: While the framework shows potential in classifying prostate cancer tumors, the generalizability of the approach to other medical imaging tasks and datasets needs further exploration.
Clarity: Some sections of the paper could benefit from additional clarity and detail, especially regarding the methodology and experimental setup.

---

### Decision · Program_Chairs · 2024-04-26

Accept